# First Detection of *Photobacterium* spp. in Acute Hemorrhagic Septicemia from the Nursehound Shark *Scyliorhinus stellaris*

**Gaetano Catanese** [1,2,*] and **Amalia Grau** [1,2]

1 Laboratorio de Investigaciones Marinas y Acuicultura (LIMIA-IRFAP), Govern de les Illes Balears, Av. Gabriel Roca 69, 07157 Port d'Andratx, Spain
2 INAGEA (UIB), Carretera de Valldemossa, km 7.5, 07122 Palma, Spain
* Correspondence: gcatanese@dgpesca.caib.es

**Abstract:** The nursehound *Scyliorhinus stellaris* is a threatened shark species and its population in the Mediterranean Sea is declining. Programs for captive breeding and repopulation in marine protected areas (MPA) are being carried out. Unfortunately, pathogens may hinder conservation plans for this species. An impactful disease of marine animals, caused by the bacteria *Photobacterium damselae*, has been detected with increased frequency in recent decades in both farmed and marine animals. The aim of this work was to determine the cause of a disease outbreak in eight captive nursehounds that died after 18 months of captivity. Gross necropsy observations were indicative of a presumptive diagnosis of hemorrhagic septicemia. Histological and molecular techniques were performed, to diagnose the etiological agents that could be involved in their mortality. Phylogenetic analysis indicated the presence of *P. damselae*, identified as subsp. *damselae* by PCR-duplex, and *Photobacterium swingsii* in the analyzed captive nursehound *Scyliorhinus stellaris*.

**Keywords:** shark; nursehound; *Scyliorhinus stellaris*; pasteurellosis; *Photobacterium*





## 1. Introduction

The nursehound *Scyliorhinus stellaris* is an elasmobranch belonging to the Scyliorhinidae family, order Carcharhiniformes, inhabiting the continental shelf over rocky and algal-covered bottoms. Usually, it is found at depths between 20 and 60 m, and up to 500 m. This shark is widespread in the whole Mediterranean, as well as in the coastal waters of the eastern Atlantic Ocean, from Morocco to the North Sea on the Scandinavian coasts [1].

The nursehound, due to overfishing and/or bycatch, which has substantially caused the decline of its population in the Mediterranean Sea [2], is currently listed by the IUCN as a "Vulnerable" species. In the Balearic Islands, it has been listed as an "Endangered" species under criteria one (I, IV) of the Balearic Red List of Fishes, as a reduction in 50% of its biomass in the last/next 10 years has been inferred by direct observation and taking into account the actual stock exploitation level [3]. The strong reduction of *S. stellaris* populations has led to the implementation of conservation plans through captive breeding and repopulation in marine protected areas (MPAs) of the Mediterranean Sea and, particularly, in the Balearic Islands (https://www.mallorcapreservation.org/grants/__trashed/, accessed on 18 December 2022), where studies on genetic diversity from different locations have begun [4].

The decline of some fish populations, including sharks, has often been associated with bycatch, the increase of pathogens in marine water, such as parasites, bacteria, and viruses, or the increase of susceptibility of the contaminated species due to pollution [5,6]. Among the emerging diseases recently detected in marine and freshwater environments, Photobacteriosis, also known as Pasteurellosis, is an infection affecting a broad range of species [7–10]. The etiological agent has been identified as *Photobacterium* spp., and particularly the *P. damselae* subspecies *damselae* and *piscicida*. These are Gram-negative,

facultatively anaerobic, motile bacteria that are pathogenic for marine animals. The two subspecies cause different diseases and only the subsp. *damselae* is zoonotic [10–15].

The two subspecies are different in biochemical phenotype, and they show different clinical signs of diseases. The *P. damselae* subsp. *piscicida*, (formerly known as *Pasteurella piscicida*) is the causative agent of photobacteriosis or the previously recognized disease known as "pasteurellosis", of marine fish. In fish affected by acute photobacteriosis, external signs are usually inconspicuous, except for the observation of slight petechial hemorrhages [16]. Although photobacteriosis may occur in both acute and chronic forms, the disease usually develops into septicemia [17].

Instead, *P. damselae* subsp. *damselae* (formerly *Vibrio damsela*) is the causative agent of "*Vibrio damsela* infection", a hemorrhagic septicemia with accompanying skin lesions in marine fish [18]. However, the *P. damselae* subsp. *damselae* is pathogenic for a wide variety of aquatic animals, both wild and farmed, such as fish, crustaceans, mollusks, and cetaceans. The organism is also a human pathogen causing necrotizing fasciitis and is considered a zoonotic agent [18]. Among the specific aquaculture hosts are *Dentex dentex* [19], *Pagrus auriga* [20], *Diplodus sargus* [21], *Sparus aurata* [22,23], *Scophtalmus maximus*, *Seriola quinqueradiata*, and *Dicentrarchus labrax*, among others [18,22]. Further, the *P. damselae* subsp. *damselae* has also been isolated from a variety of newly cultured marine fish species such as *Pagrus auriga*, *Pagrus pagrus*, *Diplodus sargus*, and *Argyrosomus regius* [18], and more recently in wild populations of *Sardinella aurita* and *Mullus surmuletus* [24].

While the *P. damselae* subsp. *damselae* is the more recognized member of the genus Photobacteria for causing fish disease, the genus nowadays comprises more than twenty-eight validated species, not all of which are pathogenic [25]. Among them, another species was newly described in marine organisms, *Photobacterium swingsii*, that was isolated from Pacific oysters (*Crassostrea gigas*) collected in Mexico and from crabs (*Maja brachydactyla*) in the Canary Islands (Spain) [26].

Recently, it was suggested that wild fish residing in polluted environments are more receptive to pathogenic microorganisms, particularly *P. damselae* for both subsp. *Damselae* and *piscicida* [24].

The *Photobacterium damselae* subsp. *damselae* is most likely an opportunistic pathogen. It has been associated with mortalities in wild sharks such as *Carcharhinus plumbeus* and *Squalus acanthias* [27,28], as well as in captive sharks held in commercial display aquaria. However, it and other *Photobacteria* species have also been shown to be part of the normal intestinal microflora of sharks [29]. In support of this, the central intestinal microflora of three shark species (*Carcharhinus brevipinna*, *Rhizoprionodon terraenovae*, and *Carcharhinus plumbeus*) was shown to share three closely related groups of bacterial species, with *Photobacterium sp.* dominating [30], as well as a fluctuating dominance of *Photobacterium* sp. was observed in intestinal microflora of scalloped hammerhead sharks (*Sphyrna lewini*) [31]. However, shark mortalities in commercial aquarium zebra shark *Stegostoma fasciatum* were identified to be caused by the *P. damselae* subsp. *damselae* [32] and smoothhound sharks (*Mustelus mustelus*) under stressful conditions from a Turkish marine aquarium affected by another species, namely *Photobacterium sanguinicancri* [17].

The present study is the first report describing *P. damselae* subsp. *damselae* and *Photobacterium swingsii* as responsible for hemorrhagic septicemia in captive nursehound sharks of *Scyliorhinus stellaris*.

## 2. Materials and Methods

### 2.1. Sampling and Placement in Aquarium

Between December 2020 and January 2021, a total of 9 nursehounds were captured by the Balearic professional fleet (using trammel nets and bottom trawling) in two different areas of the Balearic Sea (Spain; Northwestern Mediterranean Sea), with the aim of establishing a captive breeding stock for repopulation of marine protected areas (MPAs) with their offspring. During transport on the boat, fish were kept alive in a bucket with a continuous supply of seawater. Subsequently, the individuals were transferred to an

aquarium in Mallorca, Balearic Islands (Spain). There, the nursehounds ranging in length from 73–93.5 cm and weight between 2020–3990 g were quarantined separately at constant temperature for 1 month and then held in a maintenance system for approximately 18 months before mortality events occurred. The maintenance systems were tanks of 8000 L of capacity, equipped with a water supply from a well in a close recirculating system kept at specific salinity, temperature, degrees of carbonate hardness (dKH), nitrites ($NO_2$), nitrates ($NO_3$), ammonia (NH3), magnesium (Mg), calcium (Ca) and phosphates (PO4) levels. All these parameters were controlled daily. The water temperature varied naturally, being water drawn from a well, in the range of 19–22 °C between the winter and summer seasons.

The animals were fed every 48 h with chopped fresh fish (tuna, headless sardinella, anchovies, peeled shrimp, and peeled mussels), which is the same food that is supplied to the rest of the carnivorous fish maintained in the facility. The animals were perfectly adapted to the maintenance conditions, eating directly from the hand of the feeder, without being scared or stressed. After the daily feeding, the keepers removed the remains of the food. The cleaning of the tank filter is done weekly and the bottom of the tank was cleaned twice a week, rubbing with a scourer and later vacuuming the organic debris.

*2.2. Case History*

At the end of May 2022, two mortality events occurred in this maintenance system. The only surviving nursehound was transferred to another tank. The mortality episodes occurred on two different days, spaced 4 days apart between them.

On the first episode of mortality, 29 May, the fish had eaten normally during the day (13:00 h). However, that night watchman alerted at dawn (00:15) that the fishes were very active, swimming and changing direction rapidly, and that there was also food remains clogging the overflow. When the fish keepers arrived, only 15 min after the emergency call, they met 2 dead nursehounds and another was lethargic, swimming sideways and making loops. The affected nursehound was transferred to another independent tank and the following day it fully recovered. Dead nursehounds were immediately removed and frozen at −20 °C for subsequent necropsy. The rest of the specimens did not present anomalous behaviors. However, the reddish coloration of the skin and the flaccid, friable abdomen with the appearance of being filled with water were especially remarkable in these two dead nursehounds.

In the second mortality episode, on June 4, the remaining 6 surviving nursehounds died. Only the one that was separated from the previous ones in the first episode of mortality survived and later recovered. As in the previous mortality episode, the animals had eaten in the morning (13:00 h) and, at dusk (22:00 h), the night watchman alerted that he was observing dead specimens at the bottom of the tank. In addition, the water level was rising since there was food debris in the overflow and floating at the surface of the tank. When the keeper came urgently, he observed 5 dead nursehounds and one survivor very weak, swimming in loops. An extra supply of oxygen was administrated to the survivor nursehound, however, it died after a few hours. The nursehounds seemed to have vomited since the tank was found with floating food remains. Gills were reddish, with no signs of disease, however, it was noteworthy, as, in the previous episode, the reddish colors of the skin and the belly seemed to be full of water. The specimens were refrigerated for their subsequent necropsy, which was carried out the following day, together with the other previously frozen individuals.

Necropsies were performed at LIMIA-IRFAP (Marine and Aquaculture Research Laboratory of the Balearic Government). After gross examination, samples of the different tissues were taken, for histological and molecular purposes, being fixed in 10% buffered formalin for subsequent histological processing or frozen at −20 °C for molecular biology analyses.

No other animals living in the same aquarium facilities have been previously or subsequently affected by this mortality outbreak.

*2.3. Analytical Procedures*

Samples of the liver, spleen, gonad, digestive tract (stomach and anterior intestine), kidney, and gill were processed for routine histological examination. Sections were stained with Mayer's hematoxylin and eosin (MHE). Some additional sections were stained with the Brown and Brenn Gram and Ziehl-Neelsen (ZN) staining procedures to facilitate the detection of bacteria and acid-fast bacteria (i.e., *Mycobacterium* sp., *Nocardia*, *Rhodococcus*, and other acid-fast bacteria), respectively.

The DNA extractions were carried out on nonpooled samples of liver and kidney from each individual, using the commercial Macherey-Nagel DNA Tissue extraction kit (Düren, Germany), following the manufacturer's instructions. The quality and concentration of the DNA were measured using the Nanodrop ND1000 (Thermo Scientific, Waltham, MA, USA).

PCR amplifications of a fragment of the 16S rDNA gene, using the F1/R12 universal primer pair [33,34], were carried out for each sample. PCR reactions were performed in a total volume of 20 μL containing 1 μL of genomic DNA, 10 μL of KAPA Taq Ready Mix DNA Polymerase (Kapa Biosystems, Wilmington, MA, USA), 0.4 μL (20 mM) of each primer, and water to make up the final volume. After a predenaturation period for 2 min at 94 °C the amplification protocol applied in a Biometra PCR Thermocycler (Göttingen, Germany) consisted of 40 cycles of 94 °C for 30 s, 56 °C for 20 s, 72 °C for 1.30 min.

To identify the subspecies of *P. damselae*, further PCR reactions were performed in a duplex, using the primers Ure-5'/Ure-3', specific for subsp. *damselae* [35] and the primers 76a/76b [36], for amplification of subsp. *piscicida*, following the amplification protocol: 95 °C for 4 min and then 30 cycles at 95 °C for 1 min, 65 °C for 30 s and 72 °C for 1 min, with a final extension step of 5 min at 72 °C [36]. A negative amplification control was used for each PCR reaction. The amplified PCR fragments were separated in 1.5% agarose gel and stained with GelRed® Nucleic Acid Gel Stain (Biotium, Fremont, CA, USA). Amplicons were purified using a mi-PCR purification Kit (Metabion International, Planegg, Germany) following the manufacturer's instructions and sequenced in both directions using the ABI 3130 Genetic Analyzer (Applied Biosystems, Waltham, MA, USA).

Finally, the sequences were aligned to carry out a comparative analysis with known sequences present in GenBank using the BLAST application. Phylogenetic relationships were assessed using the neighbor-joining (NJ), maximum likelihood (ML) methods with MEGA X software (1000 bootstrap replicates; [37]) and Bayesian inference (BI), using MrBayes v. 3.2 (with 10,000 replicates; [38]), where every clade was supported by posterior probabilities. The sequences of some Vibrionaceae taxa available in GenBank were included in the phylogenetic analysis, using *Paraphotobacterium marinum* as an outgroup.

## 3. Results and Discussion

### 3.1. Physical and Chemical Parameters

In the aquaria, there were no abrupt changes in water temperature (ranging between 22 °C and 22.4 °C) during the time of the mortality events, as well as salinity (40%) and ph (7.78). The concentrations of Mg (1300 ppm), Ca (450 ppm), dkH (2.90 mEq/L), $NO_2$ (0 ppm), $NO_3$ (0–2 ppm), $NH_3$ (0 ppm), and $PO_4$ (0 ppm) were recorded in the tanks the same days of the death of the animals. In an aquarium, poor water quality can cause harm for fish if not maintained properly and balanced carefully. In fact, fish can become very stressed which weakens their immune system making them more susceptible to infection [39]. However, all the physical and chemical parameters of the water measured in this facility can be considered within the range of good water quality.

### 3.2. Gross Macroscopic Observations

At necropsy, disease indications were not specific, though they were indicative of acute hemorrhagic septicemia. They were consistently seen in all eight affected fish. Externally, severe dermal petechial and ecchymotic hemorrhages were observed, especially around the mouth and along the ventrum, together with abnormal abdominal swelling (Figure 1A).

Internally, a reddish serosanguinous abdominal fluid was observed, suggesting hemolysis [40]. The liver and spleen were congested, and the gastrointestinal tract was empty and covered with mucus. The observation of distended gallbladders, filled with greenish bile, was indicative of anorexia. The large ovarian congestion was one of the most striking findings (Figure 1B). No other remarkable visceral lesions were observed.

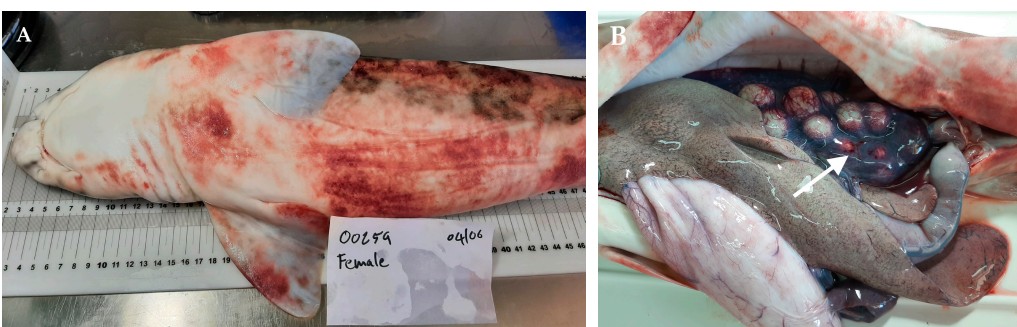

**Figure 1.** Necropsy and macroscopic lesions detected in *S. stellaris* affected individuals. (**A**) Petechial lesions and ecchymoses are mainly located on the skin of the belly and pectoral fins. (**B**) Note the intense ovarian congestion (white arrow) in another affected specimen.

### 3.3. Histopathological and Molecular Biology Analysis

The histopathological examination was only made in six of the eight affected sharks, corresponding to those that were kept fresh. The livers of the specimens presented fatty liver disease (steatosis), as occurred in most species living in captivity [41]. However, it must be taken into account that sharks do not have a swim bladder and are dependent on the lipids stored in the liver for buoyancy. Hence, this may not be the hepatic lipidosis associated with diets of captive fish [1].

Depending on the observed individual, moderate to marked autolysis was present in the tissue sections examined. Only a few significant microscopic lesions were evident and were limited to liver, spleen, and gill congestion, regardless of the position in which the specimens were stored. Hepatic or splenic granulomas were absent. Clouds of amorphous basophilic material were observed in the lumen of the renal tubules of all nursehound individuals (Figure 2), inside which Gram staining revealed isolated and weakly stained pleomorphic Gram-negative coccobacilli. No acid-fast bacteria were seen in the liver, kidney, and gills. Rafts of sloughed intestinal epithelium that contained both Gram-positive and Gram-negative bacteria were present in the intestinal lumina of all fish. Only one fish, corresponding to the one with the highest degree of autolysis, also showed pleomorphic acid-fast bacteria in its intestinal lumen.

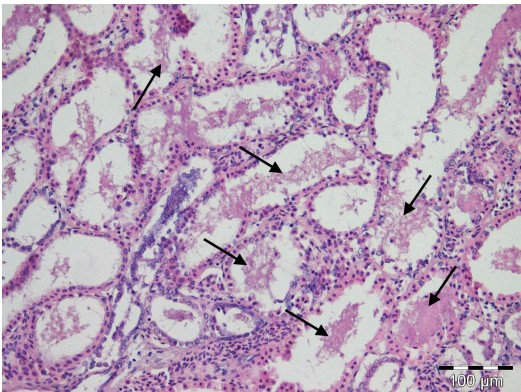

**Figure 2.** Light micrograph of a histological section through the kidney of a nursehound shark *Scylliorhinus stellaris* showing deposits of a basophilic amorphous material (black arrows) in the lumen of renal tubules. MHE staining.

Fragments of the partial 16S rDNA gene (approximately 1400 bp) were obtained from all specimens. The sequence of a single bacterium was obtained from each infected individual. Among all, only two clearly different strains were detected and the sequences were deposited in GenBank under accession numbers LC744198 and LC744199. Strain one was obtained from all samples except in one specimen; on the contrary, the sequence of strain two was obtained from only this last individual.

The BLAST result of the two strain sequences obtained from the Balearic aquarium indicated similarity (99.9–100% identity) with two types of bacteria of the same genus, *Photobacterium damselae* and *Photobacterium swingsii*. Likewise, the phylogenetic analyses unambiguously included the obtained two strain sequences within the cluster of *Photobacterium* species, supported by significant bootstrap values (Figure 3). A consistent monophyletic clade (high significant bootstrap values/Bayesian probability: 100/100/1) showed that the strain one sequence was more closely related to *P. damselae* species, although the subspecies could not be identified, while the strain two sequence was related to *P. swingsii* (significant bootstrap values/Bayesian probability: 82/71/0.79).

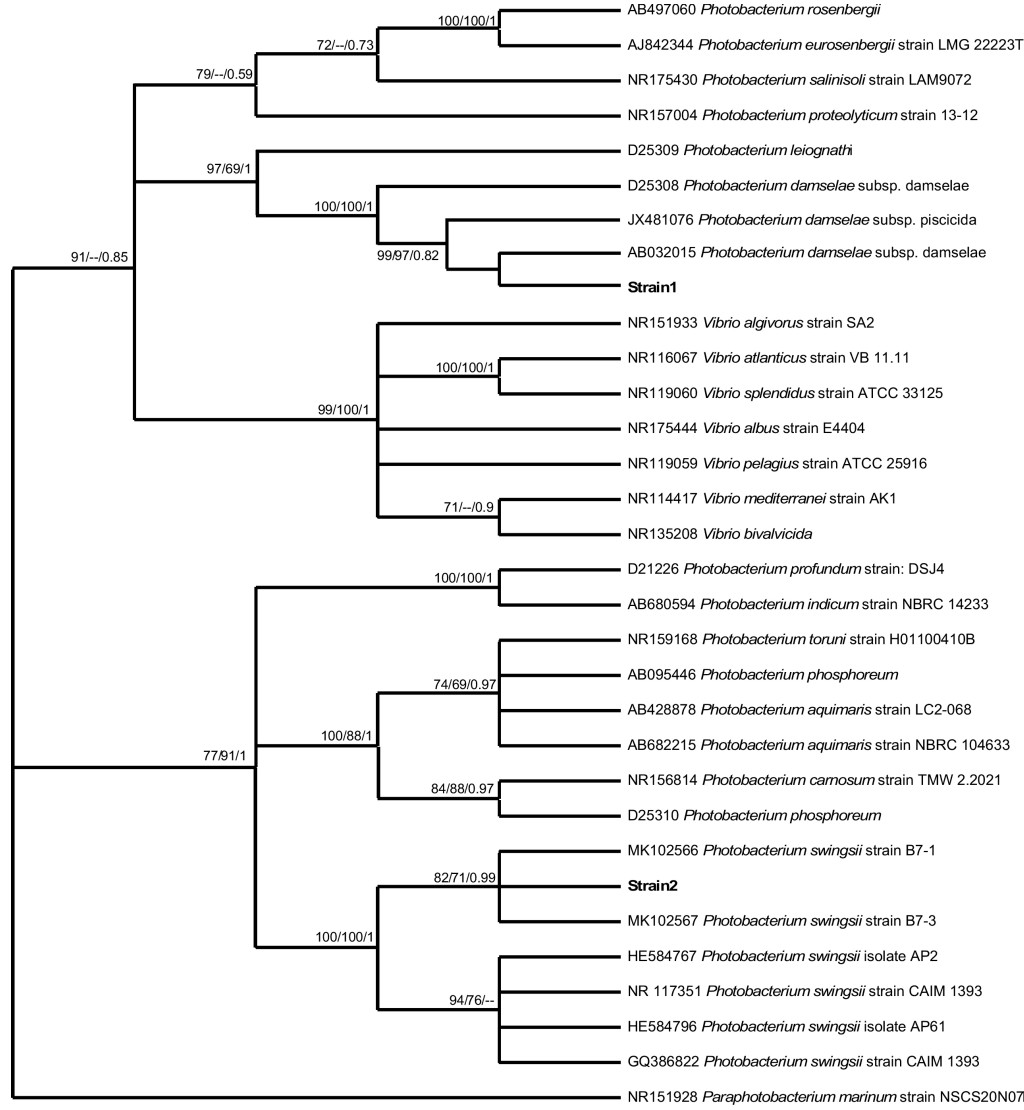

**Figure 3.** Phylogenetic relationships based on the sequence of the partial 16SrDNA gene. Neighbor-joining, maximum likelihood, and Bayesian Inference bootstrap values higher than 50% are indicated below nodes, respectively.

Moreover, the PCRs applied for the detection of *P. damselae* subspecies allowed their identification in all the seven samples in which strain one was shown with *a* partial 16S rDNA gene. The PCRs revealed only the amplification for the UreC partial gene with products of 448 bp specific for *P. damselae* subsp. *damselae* (Figure 4). On the contrary, no amplicons of 297 bp (using the primers 76a/76b), specific for *P. damselae* subsp. *Piscicida* were obtained (Figure 4).

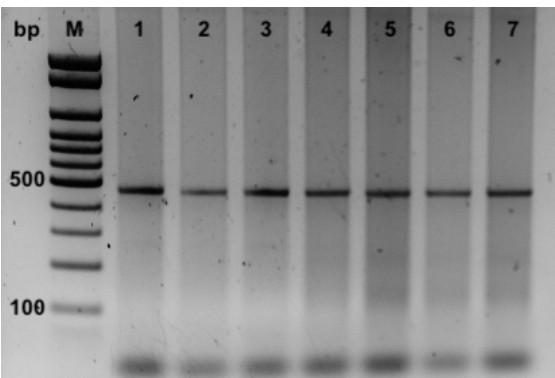

**Figure 4.** PCR products from DNA of samples of kidney for *P. damselae* subspecies identification using 76a/76b and Ure-5′/Ure-3′ primer pairs. M, 100 bp ladder; 1–7 ID of the samples.

All the analyzed nursehounds showed macroscopic pathological findings consistent with acute *Photobacterium* septicemia, similar to those described in other studies affecting different cultured species [22,42–45]. Generally, clinical signs and gross lesions in fish acutely infected with *Photobacterium* sp. may be absent and/or nonspecific. However, white nodules are often present in multiple visceral organs with the chronic form of the disease. Hence it is sometimes referred to as pseudotuberculosis [11,42]. Although viral infections cannot be ruled out, the detection of *Photobacterium* spp. makes it considered to be the most plausible causative agent of the infection.

In recent years, mortality outbreaks caused by pathogens, affecting different fish species, have been observed in Balearic waters [46–49], some of them affecting mainly MPAs [50,51]. However, no massive mortality events (MMEs) of *S. stellaris* have been observed until now in Balearic MPA.

The origin of the transmission and the presence of these pathogens in a facility is not easy to identify and different hypotheses can be proposed. In general, many wild fish do not easily adapt to artificial conditions, and when placed in aquaria, many disease problems can arise [52]. Pasteurellosis is usually related to moderate seawater temperatures, above 20 °C [53]. Below this temperature, the fish can harbor the pathogen as a subclinical infection and become a carrier for a long period of time [54]. However, considering that the nursehounds analyzed in this work were housed in aquaria for more than one year (including one when water temperatures were higher) without showing any signs of disease, hence the likelihood that the sharks were subclinically infected and carriers are relatively low. However, it cannot be discarded that the bacteria may have been present at the time of capture as part of the normal intestinal flora [29–31] and that a stressful event, connected to aquarium keeping [55–57], induced the overt infection.

Similarly, we can reject the other hypothesis of the introduction of the pathogen directly by water, due to poor filtration, because fish hosted simultaneously in different tanks would have been affected and died.

A more plausible explanation would be the transmission through food (contaminated fresh or frozen fish) or through human management operations in the aquarium facilities (for example by contaminated suits of divers). However, other animals fed the same food showed no signs of illness. Further studies, supported by complementary and more specific techniques, should be carried out to corroborate these different hypotheses.

The detection in a dead nursehound of *P. swingsii*, which is a species only recently described and poorly known, unlike other photobacteria, could indicate the character of an opportunistic pathogen of this bacterium that causes disease only under certain conditions. Conversely, the characteristics of the *P. damselae* subsp. *damselae* are known. It behaves as a generalist free-swimming bacterium and as a pathogen with hemolytic and cytolytic activities, causing disease in a wide range of animal phyla [58].

In fact, one of the clearest differences between the *P. damselae* subsp. *Damselae,* with respect to the subsp. *piscicida*, is the ability to produce hemolysis in different types of blood, both fish and mammals. Erythrocytes from different species are sensitive to both the cells and extracellular products of the *P. damselae* subsp. *damselae* [59]. The symptoms that this bacterium causes in fish are clearly related to the production of extracellular compounds that include phospholipase or hemolytic activities [60]. The hemolysis caused by the *P. damselae* subsp. *damselae* seems to be a consequence of a toxin and/or additional hemolysins [61]. In addition, various studies have shown in culture plates a clear distinction between strongly hemolytic strains and weakly hemolytic strains within the *P. damselae* subsp. *damselae* [22,35,62]. However, damages to organs and tissues that can lead to death in animals are directly related to the action of these toxins that substantially contribute to bacterial pathogenicity.

Nevertheless, it was observed that these bacteria can be transmitted, not only to other aquatic animals in the aquarium, [40] but also in the wild. Animal reintroduction from aquarium facilities into the wild must be done with great care to prevent the possible spread of disease into the natural environment. It could compromise conservation effectiveness, particularly in marine reserves [63]. In general, in controls prior to fish stocking in the wild, the general state of health of the individuals, their skin, food condition, the appearance of the gills, and everything that can determine the better physical conditions to adapt their life in freedom and the integration into the natural environment, are inspected. In addition, the main diseases of bacterial, parasitic, and fungal origin are commonly analyzed, only allowing repopulation when the absence of detection of infectious diseases which could represent a risk, is established. For this reason, in the management plans for the reintroduction of *S. stellaris*, or other shark species, in the wild and in MPAs, similarly, more controls should be carried out both immediately after their capture and routinely during captivity, making accurate observations of the general state of the individuals, and also frequent control laboratory analyses on animals, water, and even on the food that is supplied. Furthermore, adequate sanitary protocols in relation to human maintenance activities, equipment, and animal handling, will help minimize the stress of captive individuals and limit the possible entries of infection.

Therefore, particular attention should be directed towards the protection and preservation of MPAs from possible infections, avoiding their inadvertent contamination through the introduction of those microbial organisms, to date not considered particularly pathogenic for a particular species.

In recent decades, different vaccine formulations have been developed for some aqua-cultured fish species against photobacteriosis and some vaccines are already commercially available [64,65]. Therefore, further research into the development and application of new vaccines for this species of shark, including effective vaccination strategies in large display aquaria to prevent and control this disease should be incorporated into the species conservation.

The results of this work suggest paying more attention during the stabling phases and the handling of this species for conservation purposes and helping to adapt prevention and protection actions in the wild and during reintroduction into MPAs.

## 4. Conclusions

This study was the first to describe the detection of *Photobacterium damselae* and *P. swingsii* in the nursehound shark *Scyliorhinus stellaris*. The subspecies of *P. damselae* subsp. *damselae* was likely associated with hemorrhagic septicemia in the captive individuals

studied. This work was carried out using histological and molecular techniques for the identification of these bacteria. Two different PCRs were performed for preliminary identification of *P. damselae* and then to identify the subspecies of the samples found to be positive.

Although this species studied is not at imminent risk of extinction, plans for fish stocking are starting up.

Further investigations are needed for understanding the impact of this potential pathogen in the reintroduction of shark specimens in marine protected areas.

**Author Contributions:** Conceptualization, G.C. and A.G.; methodology, G.C. and A.G.; software, G.C.; investigation, G.C. and A.G.; writing—original draft preparation, G.C.; writing—review and editing, G.C. and A.G. All authors have read and agreed to the published version of the manuscript.

**Funding:** This research received no external funding.

**Institutional Review Board Statement:** Ethics Committee or Institutional Review Board approval is not required for our manuscript since the research carried out was based on the detection of the pathogens that have caused the death of the studied specimens and no experimentation with live animals or specific sacrifices for research has been done. In summary, the individuals analyzed are not animals used for experimentation, and therefore no approval from the ethics committee is necessary.

**Data Availability Statement:** The data that support the findings of this study are available on GenBank accession numbers: LC744198 and LC744199.

**Acknowledgments:** The authors express their gratitude to Dirección General de Pesca of the Consellería Agricultura, Pesca i Alimentació (Govern de les Illes Balears).

**Conflicts of Interest:** The authors declare no conflict of interest.

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
