# Peer review of "First Detection of Photobacterium spp. in Acute Hemorrhagic Septicemia from the Nursehound Shark Scyliorhinus stellaris"

_fishes, doi:10.3390/fishes8030128_

Round 1
Reviewer 1 Report (Previous Reviewer 1)
Dear authors
This time I just have minor annotations.
Throughout the document - Pay attention to the terms "rRNA" and "rDNA". Use only one of the terms.
Lines 224-226 - It is important to use a reference to support the natural presence of P.damselae in the sharks digestive tract.
Author Response
Comment: Throughout the document - Pay attention to the terms "rRNA" and "rDNA". Use only one of the terms.
Response: We changed them accordingly
Comment: Lines 224-226 - It is important to use a reference to support the natural presence of P.damselae in the shark digestive tract.
Response: We add some references
Reviewer 2 Report (New Reviewer)
The study conducted by Catanese and Grau, is interesting and well designed and written. Very few comments include, photos has no arrows as written in their caption

Author Response
Comment: The study described P. damselae subsp. damselae and Photobacterium swingsii responsible for a haemorrhagic septicaemia in stabled nursehound sharks Scyliorhinus stellaris as a first report. It address a specific gap in the field.
As an emerging disease that affects the decline in the spp and affects their conservation and also as recorded for the first time
The methodology is fine, and the authors mentioned vaccine control in their manuscript
The conclusions are consistent with the evidence and arguments presented. They addressed the main question posed.
The references are appropriate.
Additional comments: Pathology figures have no arrows so please revise.
Response: We sincerely appreciate the reviewer’s comments. We put the arrows in the figure.
Reviewer 3 Report (New Reviewer)
This is an interesting manuscript describing acute photobacteriosis in captive nursehound sharks. It is a fairly straightforward manuscript although I did wonder why normal bacterial cultures were not attempted as that would be typically done in a diagnostic workup.
While I am loathe to change the writing style of the authors, I have made some suggestions below that I hope are helpful in trying to make it more concise and accurate. I hope I have not inadvertently changed what the authors are trying to convey in interpreting what they have written and apologize in advance if I have done so.
Line 15 Suggest changing “stabled” to “captive” throughout the manuscript as I believe that that is the more commony/typical usage – stabled connotes being held in a stable.
Line 13 and 14 Suggest “… caused by the bacteria Photobacterium damselae has been detected with increased frequency in recent decades in both farmed and marine animals.”
Line 39-40 Suggest “…. where studies on genetic diversity from different locations have begun.”
Line 46-51 suggest “ …. as Photobacterium spp. particularly P. damselae subspecies damselae and piscicida. These are Gram negative, facultative anaerobic, motile bacteria that are pathogenic for marine animals. The two subspecies causes different diseases and only the subsp. damselae is zoonotic”. Suggest deleting the definition of zoonotic.
Line 52-53 I am unsure if inconspicuous nature of lesions is correct. Please see https://virtuallearn.wpengine.com/fhs/wp-content/uploads/sites/30/2017/08/1.2.14-Photobacteriosis_2014.pdf which is from the American Fisheries Society Fish Health Section blue book. Even the authors’ cited reference mentions pale gills. Do the authors mean that the lesions are minimal and non-specific rather than inconspicuous or do they want to delve into acute and chronic and the paucity of lesions in the acute form of the disease.”
Line 58-60 The way that this paragraph is currently worded may suggest/imply that the 28 species are pathogenic which I do believe they are not. May I suggest “While P. damselae subsp. damselae is the more recognized member of the genus Photobacteria for causing fish disease, the genus nowadays comprises more than twenty-eight validated species not all of which are pathogenic.”
The paragraph Line 67-78 is a little convoluted with the main point to be conveyed is that Photobacteria spp are likely to opportunistic pathogens. May I suggest something like “ Photobacterium damselae subsp. damselae is most likely an opportunistic pathogen. It has been associated with mortalities in wild sharks such as Carcharhinus plumbeus and Squalus acanthias as well as in captive sharks held in commercial display aquaria. However, it and other Photobacteria species have also been shown to be part of the normal intestinal microflora of sharks.”
Line 89-94– suggest “…they were transferred to an aquarium in Mallorca Balearic Island (Spain). There they were quarantined at constant temperature for 1 month then held in a maintenance system for approximately 18 months before mortality events occurred. Both of the quarantine and maintenance systems were close recirculating systems kept at specific salinity, temperature, nitrites, nitrates, ammonia and phosphates levels.” (It might be nice to provide the range for these parameters)
Line 96-97 - I am not totally sure what the authors are trying to convey with “oscillation rank” . Do they mean that “..the water temperature was held between 19-22C which is the maximum winter and minimum summer water temperatures of the Balearic Sea.” Please clarify.
Line 100- 106 – Suggest “At the end of May 2022, there were 2 mortality events that occurred 4 days apart. Eight nursehounds ranging in length from 73-93.5 cm and weight between 2020-3990 g were recovered. The sole surviving nursehound was transferred to another tank. The 2 dead nursehounds that were removed on May 29 were frozen at ___C while the remaining 6 sharks found dead on June 4 were refrigerated. Necropsies were performed ___ days later at LIMIA-IRFAP (Marine and Aquaculture Research Laboratory of the Balearic Government).”
Line 110 – suggest “….frozen at __C for molecular analysis.”
Line 112 – can delete “from each block”
Line 114 may want to state the town and country for Scharlab histo kit; same for line 117 Macherey-Nagel DNA Tissue extraction kit. Wanted to check on the name of the Gram stain kit as I could not find a Gram stain on their web – I did find the individual chemicals for the Gram stain. Perhaps it might be best to leave Scharlb out as the source since the source for the Hematoxylin and Eosin and the Ziehl Neelsen stains were also not specified and just state what procedure was used for Gram stain (e.g. modified Brown and Brenn method). For the DNA extraction kit was it the “NucleoSpin Tissue Mini Kit (Marcherey-Nagel, _____(town), ____ (country)”
Line 116 suggest “DNA extractions were carried out on non-pooled samples of liver and kidney from each individual ……”
Line 118 suggest “… following manufacturer’s instructions.”
Line 119 can delete “instrument” and may state the town and country for Thermo Scientific. Will need to do the same for other companies cited in the manuscript.
Line 126 – I would suggest deleting “In addition”.
Line 128 suggest “… for subsp. piscicida following the amplification protocol: 95°C for 4 min……”
Line 141 143 suggest “……in GenBank and Paraphotobacterium marinum were used as an outgroup and were included in the phylogenetic analysis.”
Line 146-148 suggest “ There were no abrupt changes water temperature during the time of the mortality events.”
Line158-154. The gross descriptions of the lesions is unfortunately are not very good and there may be improper/incorrect use of terminology. While the dermal hemorrhage is clearly evident in Figure 1A and the ovarian congestion in Figure 1B (Please also note the arrow mentioned in the caption is not evident in the photograph), the hepatic and splenic congestion as well as the peritoneal hemorrhage is not clearly visible (For hepatic congestion, I would at least expect to see a reticular pattern on the capsular surface of the liver due to blood in the sinusoids). I was unsure if the authors were indicating there was serosanguinous ascites when they indicate that there was extravasation of blood and if they meant that the plasma rather than blood was yellow. Did the authors bleed the fish to get blood to make that evaluation of yellow or was it the blood/fluid in the coelomic cavity? If it was the latter, the authors may want to take into account the delayed time to necropsy and if there was bile imbibition that may have contributed to the color of the ascites. Also, were the lesions present consistently seen in all 8 fish? Some suggestions for the description may be “Externally, there were severe dermal petechial and ecchymotic hemorrhages especially around the mouth and along the ventrum. Internally, the liver and spleen were congested and the gastrointestinal tract was empty and covered in mucus……….”
Line 160 - Unfortunately, the histopathologial descriptions are a little weak. I do need to caution the authors on their interpretation of the liver histology. Sharks do not have a swim bladder and are dependent on lipid stored in the liver for buoyancy. Hence, this may not be hepatic lipidosis associated with diets of captive fish. I am also concern about the interpretation of hepatic and splenic congestion – where were the samples taken – was it on the dependent side – i.e. was there pooling of blood in those organs (and gills) on the side the fish was laying on as the necropsy was delayed several days. Unfortunately, the renal histomicrograph is not very illuminating and probably should be left out (are there better sections that could better convey what the authors report?) unless there were sections with foci of necrosis or an inflammatory cell infiltrate in the renal interstitium that would support a bacteremia/septicemia especially where the Gram negative bacteria were found. Also please note again the arrows indicated in the legend were not evident. A Giemsa stained section may help highlight the bacteria and is sometimes used in conjunction with Gram stains (i.e. Giemsa will stain the bacteria blue and then the Gram stain of the serial section would indicate if it is Gram positive or negative). If I read the descriptions correctly, I suggest “Moderate to marked autolysis was present in the tissue sections examined. Only a few significant microscopic lesions were evident and were limited to …………(name the organs and the lesions). Hepatic or splenic granulomas were absent. Gram stains revealed aggregates(? I am not too sure if these were aggregates, individual bacteria or both) of weakly stained pleomorphic Gram-negative coccobacilli in the renal interstitium(? Not too sure where in the kidney the bacteria were seen i.e. tubules, glomeruli or interstitial tissues). No acid-fast bacteria were seen in liver, kidney and gills. Rafts of sloughed intestinal epithelium that contained both Gram positive and Gram negative bacteria.were present in the intestinal lumina of all fish. One fish also had pleomorphic acid-fast bacteria in its intestinal lumen.” I left off the amorphous basophilic material present in renal tubules and intestine as I suspect that this is part of autolysis – nuclear debris of slough cells but I was not sure from what I could see in the histomicrograph provided. May also want to state if the histopathologic findings wee similar in all 8 sharks. In that regards, was histopathologic evaluation also performed on the frozen fish and how much freeze artifact was present.
Line 202-203 – Suggest “Generally, clinical signs and gross lesions in fish acutely infected with Photobacterium sp. may be absent and/or are non-specific. However, white nodules are often present in multiple visceral organs with the chronic form of the disease hence it is some times referred to as pseudotuberculosis. “
Line 221 suggest “… housed in aquaria for more than one year (including one summer where water temperatures are higher) without showing any signs of disease, hence the likelihood that the sharks were subclinically infected and carriers is relatively low. However, it cannot be discounted that the bacteria may have been present at time of capture as part of the normal intestinal flora and that a stressful event induced the overt infection.” I worded it this way as I think it cannot be totally discounted but could be place lower on the list of possible means of transmission as there was really no possible way to prove otherwise (it would have to be sublethal sampling and sampling of the sharks when the first arrived).
Line 234 suggest… “ A more plausible explanation would be the transmission through the food (contaminated fresh or frozen fish) or through human management operations in the aquarium facilities (for example by contaminated suits of divers). However, other animals fed the same food with the same food showed no signs of illness.”
Line 248-253 suggest . “In recent decades different vaccine formulations have been developed for some aquacultured fish species against photobacteriosis and some vaccines are already commercially available [53, 54]. Therefore, further research on the development and the application of new vaccines for this species of shark including effective vaccination strategies in large display aquaria to prevent and control this disease that should be incorporated in the species conservation
Author Response
Comment: This is an interesting manuscript describing acute photobacteriosis in captive nursehound sharks. It is a fairly straightforward manuscript although I did wonder why normal bacterial cultures were not attempted as that would be typically done in a diagnostic workup.
While I am loathe to change the writing style of the authors, I have made some suggestions below that I hope are helpful in trying to make it more concise and accurate. I hope I have not inadvertently changed what the authors are trying to convey in interpreting what they have written and apologize in advance if I have done so.
Response: Thank you for your thorough review, salient observations and suggestions.
Comment: Line 15 Suggest changing “stabled” to “captive” throughout the manuscript as I believe that that is the more commony/typical usage – stabled connotes being held in a stable.
Response: We changed them accordingly
Comment: Line 13 and 14 Suggest “… caused by the bacteria Photobacterium damselae has been detected with increased frequency in recent decades in both farmed and marine animals.”
Response: We changed it accordingly
Comment: Line 39-40 Suggest “…. where studies on genetic diversity from different locations have begun.”
Response: We changed it accordingly
Comment: Line 46-51 suggest “ …. as Photobacterium spp. particularly P. damselae subspecies damselae and piscicida.These are Gram negative, facultative anaerobic, motile bacteria that are pathogenic for marine animals.The two subspecies causes different diseases and only the subsp. damselae is zoonotic”.Suggest deleting the definition of zoonotic.
Response: We changed it accordingly
Comment: Line 52-53 I am unsure if inconspicuous nature of lesions is correct. Please see https://virtuallearn.wpengine.com/fhs/wp-content/uploads/sites/30/2017/08/1.2.14-Photobacteriosis_2014.pdf which is from the American Fisheries Society Fish Health Section blue book. Even the authors’ cited reference mentions pale gills. Do the authors mean that the lesions are minimal and non-specific rather than inconspicuous or do they want to delve into acute and chronic and the paucity of lesions in the acute form of the disease.”
Response: We have exactly quoted the information that appeared in article Abu-Elala et al. 2015, where the word "inconspicuous" appeared twice, referring to the external lesions that are generally observed in fish in the acute phase of the disease. To clarify, we have further specified the meaning of the phrase and we have introduced the reference kindly provided by the referee. We have also added more bibliographical information in an attempt to clarify concepts
Comment: Line 58-60 The way that this paragraph is currently worded may suggest/imply that the 28 species are pathogenic which I do believe they are not. May I suggest “While P. damselae subsp. damselae is the more recognized member of the genus Photobacteria for causing fish disease, the genus nowadays comprises more than twenty-eight validated species not all of which are pathogenic.”
Response: We changed it accordingly
Comment: The paragraph Line 67-78 is a little convoluted with the main point to be conveyed is that Photobacteria spp are likely to opportunistic pathogens. May I suggest something like “ Photobacterium damselae subsp. damselae is most likely an opportunistic pathogen.It has been associated with mortalities in wild sharkssuch as Carcharhinus plumbeus and Squalus acanthiasas well as in captive sharks held in commercial display aquaria. However, it and other Photobacteria species have also been shown to be part of the normal intestinal microflora of sharks.”
Response: We changed it accordingly
Comment: Line 89-94– suggest “…they were transferred to an aquarium in Mallorca Balearic Island (Spain).There they were quarantined at constant temperature for 1 month then held in a maintenance system for approximately 18 months before mortality events occurred. Both of the quarantine and maintenance systems were close recirculating systems kept at specific salinity, temperature, nitrites, nitrates, ammonia and phosphates levels.” (It might be nice to provide the range for these parameters).
Response: We changed it accordingly and we provided in a new section the values of the water parameters
Comment: Line 96-97- I am not totally sure what the authors are trying to convey with “oscillation rank” . Do they mean that “..the water temperature was held between 19-22C which is the maximum winter and minimum summer water temperatures of the Balearic Sea.”Please clarify.
Response: We changed the sentence but we meant that the water temperatures varied naturally in this range between winter and summer seasons
Comment: Line 100- 106 – Suggest “At the end of May 2022, there were 2 mortality events that occurred 4 days apart. Eight nursehounds ranging in length from 73-93.5 cm and weight between 2020-3990 g were recovered.The sole surviving nursehound was transferred to another tank.The 2 dead nursehounds that were removed on May 29 were frozen at ___C while the remaining 6 sharks found dead on June 4 were refrigerated.Necropsies were performed ___ days later at LIMIA-IRFAP (Marine and Aquaculture ResearchLaboratory of the Balearic Government).”
Response: We changed it accordingly
Comment: Line 110 – suggest “….frozen at __C for molecular analysis.”
Response: We changed it accordingly
Comment: Line112 – can delete “from each block”
Response: We changed it accordingly
Comment: Line 114 may want to state the town and country for Scharlab histo kit; same for line 117 Macherey-Nagel DNA Tissue extraction kit. Wanted to check on the name of the Gram stain kit as I could not find a Gram stain on their web – I did find the individual chemicals for the Gram stain. Perhaps it might be best to leave Scharlb out as the source since the source for the Hematoxylin and Eosin and the Ziehl Neelsen stains were also not specified and just state what procedure was used for Gram stain (e.g. modified Brown and Brenn method). For the DNA extraction kit was it the “NucleoSpin Tissue Mini Kit (Marcherey-Nagel, _____(town), ____ (country)”
Response:
We changed them accordingly
Comment: Line 116 suggest “DNA extractions were carried out on non-pooled samples of liver and kidney from each individual ……”
Response: We changed it accordingly
Comment: Line 118 suggest “… following manufacturer’s instructions.”
Response: We changed it accordingly
Comment: Line 119 can delete “instrument” and may state the town and country for Thermo Scientific. Will need to do the same for other companies cited in the manuscript.
Response: We changed them accordingly
Comment: Line 126 – I would suggest deleting “In addition”.
Response: We changed it accordingly
Comment: Line 128 suggest “… for subsp. piscicida following the amplification protocol:95°C for 4 min……”
Response: We changed it accordingly
Comment: Line 141 143 suggest “……in GenBank and Paraphotobacterium marinum were used as an outgroup and were included in the phylogenetic analysis.”
Response: We changed this sentence
Comment: Line 146-148 suggest “ There were no abrupt changes water temperature during the time of the mortality events.”
Response: We changed it accordingly
Comment: Line158-154.The gross descriptions of the lesions is unfortunately are not very good and there may be improper/incorrect use ofterminology.While the dermal hemorrhage is clearly evident in Figure 1A and the ovarian congestion in Figure 1B (Please also note the arrow mentioned in the caption is not evident in the photograph), the hepatic and splenic congestion as well as the peritoneal hemorrhage is not clearly visible (For hepatic congestion, I would at least expect to see a reticular pattern on the capsular surface of the liver due to blood in the sinusoids).I was unsure if the authors were indicating there was serosanguinous ascites when they indicate that there was extravasation of blood and if they meant that the plasma rather than blood was yellow.Did the authors bleed the fish to get blood to make that evaluation of yellow or was it the blood/fluid in the coelomic cavity?If it was the latter, the authors may want to take into account the delayed time to necropsy and if there was bile imbibition that may have contributed to the color of the ascites.Also, were the lesions present consistently seen in all 8 fish? Some suggestions for the descriptionmay be “Externally, there were severe dermal petechial and ecchymotic hemorrhages especially around the mouth and along the ventrum.Internally, the liver and spleen were congested and the gastrointestinal tract was empty and covered in mucus……….”
Response: We fully agree with the reviewer that the description of gross lesions is imprecise. The manuscript was initially a brief communication and the necessary shortness, implicit in this type of manuscript, limited the extension of the macroscopic description. Based on the reviewer's suggestions, we improved the description
Comment: Line 160- Unfortunately, the histopathologial descriptions are a little weak. I do need to caution the authors on their interpretation of the liver histology. Sharks do not have a swim bladder and are dependent on lipid stored in the liver for buoyancy. Hence, this may not be hepatic lipidosis associated with diets of captive fish. I am also concern about the interpretation of hepatic and splenic congestion – where were the samples taken – was it on the dependent side – i.e. was there pooling of blood in those organs (and gills) on the side the fish was laying on as the necropsy was delayed several days. Unfortunately, the renal histomicrograph is not very illuminating and probably should be left out (are there better sections that could better convey what the authors report?) unless there were sections with foci of necrosis or an inflammatory cell infiltrate in the renal interstitium that would support a bacteremia/septicemia especially where the Gram negative bacteria were found. Also please note again the arrows indicated in the legend were not evident. A Giemsa stained section may help highlight the bacteria and is sometimes used in conjunction with Gram stains (i.e. Giemsa will stain the bacteria blue and then the Gram stain of the serial section would indicate if it is Gram positive or negative). If I read the descriptions correctly, I suggest “Moderate to marked autolysis was present in the tissue sections examined.Only a few significant microscopic lesions were evident and were limited to …………(name the organs and the lesions).Hepatic or splenic granulomas were absent. Gram stains revealed aggregates(? I am not too sure if these were aggregates, individual bacteria or both) of weakly stained pleomorphic Gram-negative coccobacilli in the renal interstitium(? Not too sure where in the kidney the bacteria were seen i.e. tubules, glomeruli or interstitial tissues). No acid-fast bacteria were seen in liver, kidney and gills. Rafts of sloughed intestinal epithelium that contained both Gram positive and Gram negative bacteria.were present in the intestinal lumina of all fish.One fish also had pleomorphic acid-fast bacteria in its intestinal lumen.”I left off the amorphous basophilic material present in renal tubules and intestine as I suspect that this is part of autolysis – nuclear debris of slough cells but I was not sure from what I could see in the histomicrograph provided. May also want to state if the histopathologic findings wee similar in all 8 sharks. In that regards, was histopathologic evaluation also performed on the frozen fish and how much freeze artifact was present.
Response: We agree again with the reviewer that the histopathological description could be improved. We changed it accordingly with the suggestions kindly provided.
Comment: Line 202-203 – Suggest “Generally, clinical signs and gross lesions in fish acutely infected with Photobacterium sp.may be absent and/or are non-specific. However, white nodules are often present in multiple visceral organs with the chronic form of the disease hence it is some times referred to as pseudotuberculosis. “
Response: We changed it accordingly
Comment: Line 221 suggest “… housed in aquaria for more than one year (including one summer where water temperatures are higher) without showing any signs of disease, hence the likelihood that the sharks were subclinically infected and carriers is relatively low.However, it cannot be discounted that the bacteria may have been present at time of capture as part of the normal intestinal flora and that a stressful event induced the overt infection.” I worded it this way as I think it cannot be totally discounted but could be place lower on the list of possible means of transmission as there was really no possible way to prove otherwise (it would have to be sublethal sampling and sampling of the sharks when the first arrived).
Response: We changed it accordingly
Comment: Line 234 suggest… “ A more plausible explanation would be the transmission through the food (contaminated fresh or frozen fish) or through human management operations in the aquarium facilities (for example by contaminated suits of divers). However, other animals fed the same food with the same food showed no signs of illness.”
Response: We changed it accordingly
Comment: Line 248-253 suggest. “In recent decades different vaccine formulations have been developed for some aquacultured fish species against photobacteriosis and some vaccines are already commercially available [53, 54]. Therefore, further research on the development and the application of new vaccines for this species of shark including effective vaccination strategies in large display aquaria to prevent and control this disease that should be incorporated in the species conservation
Response: We changed it accordingly
Reviewer 4 Report (New Reviewer)
The manuscript "First detection of Photobacterium spp. in acute haemorrhagic septicaemia from the nursehound shark Scyliorhinus stellaris" talks about the identification of two species of Photobacterium in sharks kept in the aquarium.
Although the topic is of interest and little investigated, the article has several shortcomings, especially in the methodological part, which are inevitably reflected in the whole work. The lack of culture examination and the search for non-bacterial pathogens are the major limitations of the study, which must necessarily be highlighted or implemented. For this reason, I will reconsider the article following a major revision, in which these aspects will necessarily need to be clarified
Check in the entire manuscript that scientific names are written in italics (e.g. Photobacterium in line 21).
Introduction
- Line 13: I wouldn't say Photobacterium sp. infections are emerging, as they are known conditions. Rather, I would opt for another term such as "severe" or "impactful".
- Lines 46-49: rewrite or divide the sentence to emphasize the concepts you want to express.
Materials and methods
- The authors talked about the use of biomolecular techniques for the identification of bacterial pathogens. Why was a culture not conducted? in addition, why was the focus only on the typing of P. damselae? Were other bacterial pathogens not considered?
- A serious limitation of your study is the absence of analyzes conducted for the identification of viral and parasitic pathogens. Regarding the latter, I can agree in not conducting the analyzes on frozen samples, but I believe that they should be sought on refrigerated samples (although the bibliography is scarce).
- Consider dividing paragraph 2.2 into several subsections.
- Line 115: in addition to Mycobacterium, other acid-fast bacteria should also be indicated (e.g. Nocardia, Rhodococcus).
- Line 120: specify that primers F1 and R12 are universal for the 16S rRNA gene.
- Line 130: you talked about negative control; wasn't a positive control also used instead? If yes, please specify which control was used.
Results and discussion
- Lines 160-161: not all species kept in captivity have steatosis. Eliminate the statement "the typical fatty liver of fish kept in captivity", replacing it with fish species that can present steatosis in the event of captivity (e.g. sturgeons).
- Lines 224-227: in support of your hypothesis, i.e. of stressful conditions connected to aquarium keeping, you could cite some works of infections in Elasmobranchs kept in captivity conditions:
Desoubeaux, G.; Debourgogne, A.; Wiederhold, N.P.; Zaffino, M.; Sutton, D.; Burns, R.E.; Frasca, S.; Hyatt, M.W.J.; Cray, C. Multi-locus sequence typing provides epidemiological insights for diseased sharks infected with fungi belonging to the Fusarium solani species complex, Med. Mycol., 2018, 56 (5), 591–601. https://doi.org/10.1093/mmy/myx089.
Emam, A.M.; Hashem, M.; Gadallah, A.M.; Haridy, M. An outbreak of Vibrio alginolyticus infection in aquarium-maintained dark-spotted (Himantura uarnak) and Tahitian (H. fai) stingrays. Egypt. J. Aquat. Res., 2019, 45 (2), 153-158. https://doi.org/10.1016/j.ejar.2019.05.003
Tomasoni, M.; Esposito, G.; Mugetti, D.; Pastorino, P.; Stoppani, N.; Menconi, V.; Gagliardi, F.; Corrias, I.; Pira, A.; Acutis, P.L.; Dondo, A.; Prearo, M.; Colussi, S. The Isolation of Vibrio crassostreae and V. cyclitrophicus in Lesser-Spotted Dogfish (Scyliorhinus canicula) Juveniles Reared in a Public Aquarium. J. Mar. Sci. Eng. 2022, 10, 114. https://doi.org/10.3390/jmse10010114
- Discussions should be implemented more on identified pathogens.
Author Response
The manuscript "First detection of Photobacterium spp. in acute haemorrhagic septicaemia from the nursehound shark Scyliorhinus stellaris" talks about the identification of two species of Photobacterium in sharks kept in the aquarium.
Although the topic is of interest and little investigated, the article has several shortcomings, especially in the methodological part, which are inevitably reflected in the whole work. The lack of culture examination and the search for non-bacterial pathogens are the major limitations of the study, which must necessarily be highlighted or implemented. For this reason, I will reconsider the article following a major revision, in which these aspects will necessarily need to be clarified
Check in the entire manuscript that scientific names are written in italics (e.g. Photobacterium in line 21).
Response: We changed them accordingly
Introduction
Comment:- Line 13: I wouldn't say Photobacterium sp. infections are emerging, as they are known conditions. Rather, I would opt for another term such as "severe" or "impactful".
Response: We changed it accordingly
Comment:- Lines 46-49: rewrite or divide the sentence to emphasize the concepts you want to express.
Response: We changed it accordingly
Materials and methods
Comment:- The authors talked about the use of biomolecular techniques for the identification of bacterial pathogens. Why was a culture not conducted? in addition, why was the focus only on the typing of P. damselae? Were other bacterial pathogens not considered?
Response: The aim of the paper was the first detection of Photobacterium spp. in individuals of S. stellaris, and due to the fact that these bacteria were all found in dead and injured individuals, they were associated with the event of mortality. Moreover, once the photobacterium was detected, the subspecies was typed only for greater interest and to validate the previous results.
Anyway, we have not avoided considering other pathogens in our work. As consequence of obtaining a single PCR amplification of a 16rDNA fragment for each sample, using universal primers and without observing unspecific amplicons, as well as the sequencing of a single DNA fragment, led us to suppose that other bacteria, if present, could be considered negligible. However, the reviewer is right, when he considers that there may be some species of bacteria present and that they could only be detected after a culture. We do not argue and say that further studies should be conducted. However, as we said, this was not the aim of the paper which proposed to highlight the presence of photobacterium.
Comment:- A serious limitation of your study is the absence of analyzes conducted for the identification of viral and parasitic pathogens. Regarding the latter, I can agree in not conducting the analyzes on frozen samples, but I believe that they should be sought on refrigerated samples (although the bibliography is scarce).
Response: We thank the reviewer for her/his comment, but the fact of having carried out a complete histopathological analysis of the different tissue samples, together with an exhaustive gross macropscopic evaluation, allows us to rule out the presence of macro and microscopic parasites in the affected specimens. It is true that virological analyzes have not been carried out, but the lesions observed are consistent with acute hemorrhagic septicemia whose most plausible cause is consistent with the detection of photobacterium spp, according to the available bibliography. In any case, and according to the reviewer, we add a sentence in the discussion that, although viral infections cannot be ruled out, the detection of Photobacterium makes it be considered the most plausible causative agent of causing the infection.
Comment:- Consider dividing paragraph 2.2 into several subsections.
Response: We changed the section 2, rearranging the text into different subsections
Comment:- Line 115: in addition to Mycobacterium, other acid-fast bacteria should also be indicated (e.g. Nocardia, Rhodococcus).
Response: We added them accordingly
Comment:- Line 120: specify that primers F1 and R12 are universal for the 16S rRNA gene.
Response: We specified it accordingly
Comment:- Line 130: you talked about negative control; wasn't a positive control also used instead? If yes, please specify which control was used.
Response: We used negative control to discard any possible contamination. We did not use a positive control. This sort of control is particularly useful for validating the experimental procedure. However, it was not necessary since our PCRs (for the 16S rDNA gene) did not show lack of amplifications, nor was our intention to amplify a specific fragment exclusively for P. damselae.
In relation to the duplex PCR for the identification of the subspecies, the protocol described by Amagliani et al. involves the amplification of one fragment or another (in rare cases both) to discriminate P.d subs. damselae and piscicidae. In our case, amplification was obtained in all the seven reactions, making the use of a positive control unnecessary.
Results and discussion
Comment:- Lines 160-161: not all species kept in captivity have steatosis. Eliminate the statement "the typical fatty liver of fish kept in captivity", replacing it with fish species that can present steatosis in the event of captivity (e.g. sturgeons).
Response: We eliminated it and replaced the sentence accordingly
Comment:- Lines 224-227: in support of your hypothesis, i.e. of stressful conditions connected to aquarium keeping, you could cite some works of infections in Elasmobranchs kept in captivity conditions:
Desoubeaux, G.; Debourgogne, A.; Wiederhold, N.P.; Zaffino, M.; Sutton, D.; Burns, R.E.; Frasca, S.; Hyatt, M.W.J.; Cray, C. Multi-locus sequence typing provides epidemiological insights for diseased sharks infected with fungi belonging to the Fusarium solani species complex, Med. Mycol., 2018, 56 (5), 591–601. https://doi.org/10.1093/mmy/myx089.
Emam, A.M.; Hashem, M.; Gadallah, A.M.; Haridy, M. An outbreak of Vibrio alginolyticus infection in aquarium-maintained dark-spotted (Himantura uarnak) and Tahitian (H. fai) stingrays. Egypt. J. Aquat. Res., 2019, 45 (2), 153-158. https://doi.org/10.1016/j.ejar.2019.05.003
Tomasoni, M.; Esposito, G.; Mugetti, D.; Pastorino, P.; Stoppani, N.; Menconi, V.; Gagliardi, F.; Corrias, I.; Pira, A.; Acutis, P.L.; Dondo, A.; Prearo, M.; Colussi, S. The Isolation of Vibrio crassostreae and V. cyclitrophicus in Lesser-Spotted Dogfish (Scyliorhinus canicula) Juveniles Reared in a Public Aquarium. J. Mar. Sci. Eng. 2022, 10, 114. https://doi.org/10.3390/jmse10010114
Response: We added them accordingly
Comment:- Discussions should be implemented more on identified pathogens.
Response: We added more discussion and references about the pathogen
This manuscript is a resubmission of an earlier submission. The following is a list of the peer review reports and author responses from that submission.
Round 1
Reviewer 1 Report
General observations
Introduction:
Some references are a bit old.
As a paper about conservation, you could write a bit about the importance of this shark species (or sharks in general) to the environment and its conservation.
Materials and methods:
You should write the brands of the equipments and reagents, and include the PCR conditions.
Results and discussion:
You can not say that Photobacterium damselae damselae was detected by only sequencing 16S. This gene is very similar (or even equal) between both subspecies of photobacterium damselae. To confirm this, I aligned 4 sequences of each (MK285692, MK285691, MK285690, MK285689; AY147860, MT158701, MT158700, JX481076), and such similarity was confirmed. To clearly identify photobacterium subspecies, you must sequence another gene. There are at least 2 genes able to differentiate both subspecies, "putative penicillin-binding protein (Pbp) 1A" and "urease (ureC)" gene. Besides this, it is important to include P.b.piscicida sequences in the phylogenetic tree.
If you want to identify the subspecies of strain1, it is important to sequence another gene. With the actual results, in my opinion, you can not say that strain1 is subspecies damselae.
Line 43
You could wright some species's examples affected by pasteurellosis.
Line 44
Italic
Line 45
Piscicidae? Do you mean piscicida?
Line 46
More references.
Line 46-48
One or two more references.
Line 56
Italic, and specify photobacterium subspecies.
Line 74
Registered instead of produced.
Line 75
Time gap instead of Lapse of time.
Line 78
While the remaining 6
Line 83
Which part of the digestive track? It is very long. Unspecific term.
Line 88
I don't agree with the tissues mixing for DNA extraction.
However, according to the type and aim of this work it is acceptable.
Line 89
"Was used" instead of "has been used"
Line 91
"Was measured" instead of "have been measured"
Brand of the nanodrop is missing, as well as the brands of the reagents and other equipments.
Line 92
Sequences of the primers are missing, as well as the PCR conditions.
Line 94
Were purified
Line 96
Were aligned
Line 105-107
You forgot to get this information out.
Line 115
Reference to "haemolysis"
Line 119
Steosis is fatty liver disease. Instead of "fatty hepatic steatosis", I suggest "Fatty liver disease (steatosis)"
Line 120
Are observed in fig1, or were observed
Figure 2
You could show on the figure what you observed.
Line 126-127 and line 131
These lines disagree between them.
Line 141
"bacteria of the same order". Do you mean "the same genus"?
Line 142
It is risky to admit that strain 1 is a Photobacterium damselae subs. damselae using 16S. 16S rRNA of both photobacterium damselae subspecies is very similar, it might not be specific enough. Therefore, i suggest two important things: include 16S photobacterium damselae piscicida sequences in the phylogenetic tree; and sequencing another gene, in order to clarify if strain 1 is damselae or piscicida subspecies (there is a gene that allows to differentiate these subspecies, that Carlos Osório used in one of his works).
Figure 3
As Pdpiscicida is another main causative agent of fish pasteurellosis, it is important to include photobacterium damselae subs. piscicida sequences in the phylogenetic tree. This will help to clarify the classification of strain1.
Line 152
Fish instead of fishes
Line 157
"Have been observed" instead of "Are being"
Line 158
What is MME?
Line 164
Not "º"
The
Line 166
Were instead of have been
Line 158 and 180
Please substitute the word "anyway".
Line 190-192
Sentence without connection with the previous sentence.
Author Response
As a paper about conservation, you could write a bit about the importance of this shark species (or sharks in general) to the environment and its conservation.
Materials and methods:
You should write the brands of the equipments and reagents, and include the PCR conditions.
We agree and we added new sentences
Results and discussion:
You can not say that Photobacterium damselae damselae was detected by only sequencing 16S. This gene is very similar (or even equal) between both subspecies of photobacterium damselae. To confirm this, I aligned 4 sequences of each (MK285692, MK285691, MK285690, MK285689; AY147860, MT158701, MT158700, JX481076), and such similarity was confirmed. To clearly identify photobacterium subspecies, you must sequence another gene. There are at least 2 genes able to differentiate both subspecies, "putative penicillin-binding protein (Pbp) 1A" and "urease (ureC)" gene. Besides this, it is important to include P.b.piscicida sequences in the phylogenetic tree.
If you want to identify the subspecies of strain1, it is important to sequence another gene. With the actual results, in my opinion, you can not say that strain1 is subspecies damselae.
We agree with the reviewer comment. We deleted the information of the subspecies, and we added this sentence: “However, our results allowed for the identification of species but further studies will be needed using more molecular markers to identify subspecies”. In addition we made a new phylogenetic tree including P. damselae subsp. piscicida sequence.
Line 43 You could wright some species's examples affected by pasteurellosis.
Examples of fish species affected by Photobacterium are shown in the next paragraph
Line 44 Italic
We changed it accordingly
Line 45 Piscicidae? Do you mean piscicida?
Yes, we do. We changed it accordingly.
Line 46 More references.
Line 46-48 One or two more references.
We added new references
Terceti, M. S., Ogut, H., and Osorio, C. R. (2016). Photobacterium damselae subsp. damselae, an emerging fish pathogen in the Black Sea: evidence of a multiclonal origin. Appl. Environ. Microbiol. 82, 3736–3745. doi: 10.1128/AEM.00781-16
Fouz B, Toranzo AE, Marco-Noales E, Amaro C. Survival of fish-virulent strains of Photobacterium damselae subsp. Damselae in seawater under starvation conditions. FEMS Microbiol Lett 1998, 168, 181-186.
Morris JG JR, Wilson R, Hollis DG, Weaver RE, Miller HG, Tacket CO, Hickamn FW. Illness caused by Vibrio damsela and Vibrio hollisae. Lancet 1982; 319, 1294-1297.
Yamane, K., Asato, J., Kawade, N., Takahashi, H., Kimura, B., and Arakawa, Y. Two cases of fatal necrotizing fasciitis caused by Photobacterium damsela in Japan. J. Clin. Microbiol. 2004, 42, 1370–1372. doi:10.1128/JCM.42.3.1370-1372.2004.
Nakamura, Y., Uchihira, M., Ichimiya, M., Morita, K., and Muto, M. . Necrotizing fasciitis of the leg due to Photobacterium damsela. J. Dermatol. 2008 35, 44–45. doi:10.1111/j.1346-8138. 2007.00412.x.
Aigbivbalu L, Maraqa N. Photobacterium damsela wound infection in a 14-year-old surfer. South Med J. 2009;102, 425-6. doi: 10.1097/SMJ.0b013e31819b9491.
Line 56 Italic, and specify photobacterium subspecies.
We have specified the species and subespecies of Photobacteria
Line 74 Registered instead of produced.
We changed it accordingly.
Line 75 Time gap instead of Lapse of time.
We changed it accordingly
Line 78 While the remaining 6
We changed it accordingly
Line 83 Which part of the digestive track? It is very long. Unspecific term.
We specified the part of digestive tract (stomach and anterior intestine)
Line 88 I don't agree with the tissues mixing for DNA extraction. However, according to the type and aim of this work it is acceptable.
Even though the mixing methodology could present some problems, we chose to do it as a pathogen screening
Line 89 "Was used" instead of "has been used"
We changed it accordingly
Line 91 "Was measured" instead of "have been measured"
Brand of the nanodrop is missing, as well as the brands of the reagents and other equipments.
We changed it accordingly
Line 92 Sequences of the primers are missing, as well as the PCR conditions.
Sequences of the primers are described by other authors (Dorsch and Stackebrandt, 1992; Stackebrandt and Charfreitag, 1990). We added the PCR conditions in a new sentence
Line 94 Were purified
We changed it accordingly
Line 96 Were aligned
We changed it accordingly
Line 105-107 You forgot to get this information out.
We changed it accordingly
Line 115 Reference to "haemolysis"
We add the following reference:
Rivas A.J., Lemos M.L. & Osorio C.R. (2013) Photobacterium damselae subsp. damselae, a bacterium pathogenic for marine animals and humans. Frontiers in Microbiology 4, 1–5
Line 119 Steosis is fatty liver disease. Instead of "fatty hepatic steatosis", I suggest "Fatty liver disease (steatosis)"
We changed it accordingly
Line 120 Are observed in fig1, or were observed
We changed it accordingly
Figure 2 You could show on the figure what you observed.
We changed it accordingly
Line 126-127 and line 131 These lines disagree between them.
We changed the last sentence
Line 141 “bacteria of the same order”. Do you mean “the same genus”?
Yes, we do. We changed it accordingly
Line 142 It is risky to admit that strain 1 is a Photobacterium damselae subs. damselae using 16S. 16S rRNA of both photobacterium damselae subspecies is very similar, it might not be specific enough. Therefore, i suggest two important things: include 16S photobacterium damselae piscicida sequences in the phylogenetic tree; and sequencing another gene, in order to clarify if strain 1 is damselae or piscicida subspecies (there is a gene that allows to differentiate these subspecies, that Carlos Osório used in one of his works).
We agree and we added a new sentence
Figure 3 As Pdpiscicida is another main causative agent of fish pasteurellosis, it is important to include photobacterium damselae subs. piscicida sequences in the phylogenetic tree. This will help to clarify the classification of strain1.
We agree with the reviewer comment. We made a new phylogenetic tree (including P. damselae subsp. piscicida sequence) and we added new sentences in the text.
Line 152 Fish instead of fishes
We changed it accordingly
Line 157 “Have been observed” instead of “Are being”
We changed it accordingly
Line 158 What is MME?
OK, we added more information
Line 164 Not "º" The
We changed it accordingly
Line 166 Were instead of have been
We changed it accordingly
Line 158 and 180 Please substitute the word "anyway".
We changed them accordingly
Line 190-192 Sentence without connection with the previous sentence.
We changed the last sentence.

Reviewer 2 Report
Plagiarism: 21%
Overall comments
The paper describes a host-pathogen interaction of the presence of photobacterium species in septicaemia of nursehound shark. The paper has three aspects: the disease diagnosis; bacterial identification + phylogeny and histopathology of infected tissue samples. The work is concise, significant and informative. However, in my opinion, the paper has the following shortcomings.
Specific comments
1. The methods and materials section needs subheadings to increase the credibility of the paper. I am of the understanding that the shark capture was not done by the authors but by a different body. But given to the hypothesis that the authors have made in discussion, I suggest the authors to mention vividly the sampling & acclimatization phase for better understanding as it has direct impact on disease progression.
2. In results section, when temperature is a differentiating factor, please provide the data of before and after disease progression. Moreover, Figure 2 needs proper markings, also figures supporting histopathological data for organs other than kidney are missing.
3. The primers used (F1/R12) are universal primers. In this case, the authors should clearly mention whether a single bacterium was obtained from infected tissues or a particular bacteria was targeted? If latter is the case, please mention clearly the culture procedure. Additionally, in this regard, the genebank accession numbers are not found in NCBI database is it yet to be published, please confirm. Please mention clearly about which strain/bacteria obtained from which organ or one from all organ, whichever be the case.
4. The probable source of the bacterial infection could further enhance the purpose of the paper.
5. The minor revisions are mentioned in the pdf, marked as highlighted with comments added.
Suggested: Major revision

Author Response
Overall comments
The paper describes a host-pathogen interaction of the presence of photobacterium species in septicaemia of nursehound shark. The paper has three aspects: the disease diagnosis; bacterial identification + phylogeny and histopathology of infected tissue samples. The work is concise, significant and informative. However, in my opinion, the paper has the following shortcomings.
Specific comments
- The methods and materials section needs subheadings to increase the credibility of the paper. I am of the understanding that the shark capture was not done by the authors but by a different body. But given to the hypothesis that the authors have made in discussion, I suggest the authors to mention vividly the sampling & acclimatization phase for better understanding as it has direct impact on disease progression.
We added subheadings in the M&M section and we have changed different sentences accordingly.
- In results section, when temperature is a differentiating factor, please provide the data of before and after disease progression. Moreover, Figure 2 needs proper markings, also figures supporting histopathological data for organs other than kidney are missing.
We have added in material and methods section the oscillation temperature rank along the year. In addition, we have added in results more information about temperatures and we have added proper markings in Figure 1 and 2.
As it was an acute process, we have not observed relevant histopathological lesions in the other organs than kidney. For this reason, we have not included photomicrographs corresponding to the other organs to prioritize the brevity of the manuscript.
- The primers used (F1/R12) are universal primers. In this case, the authors should clearly mention whether a single bacterium was obtained from infected tissues or a particular bacteria was targeted? If latter is the case, please mention clearly the culture procedure. Additionally, in this regard, the genbank accession numbers are not found in NCBI database is it yet to be published, please confirm. Please mention clearly about which strain/bacteria obtained from which organ or one from all organ, whichever be the case.
We added that a single bacterium was obtained from each individual PCR (from a mixture of tissues as indicated in M&M), according with the reviewer suggestion. Moreover, we added the sentence “A mixture of different tissues from each sample (liver, spleen, digestive tract, kidney) was used for DNA extraction of each individual”
The accession to NCBI database GenBank will be opened and made public on February 1st, 2023.
- The probable source of the bacterial infection could further enhance the purpose of the paper.
We agree with the reviewer’s comment but at the moment we can only make general hypotheses with greater or lesser probability, without having any evidence of the source of the infection
- The minor revisions are mentioned in the pdf, marked as highlighted with comments added.
We changed them accordingly
Reviewer 3 Report
The paper is interesting and well written. My only concern is that there is no proof that these bacteria are cause of death of these animals.
The authors should be more cautious about the conclusions. For example line 17:
"Histological and molecular techniques were performed, to diagnose the aetiological agents involved in their mortality."
"Involved" is too strong. The authors should add "could be involved".
Minor editing problems:
line 19: indicated the presence of P. damselae subsp damselae y Photobacterium swingsii in the analyzed
Subspecies name must be italicized and change "y" by "and".
Line 33: criteria 1 (I, IV)
Please explain because UICN red list criteria are not indicated like this:
See https://portals.iucn.org/library/node/7977
Line 40: with the increase of pathogens in marine water,
It can be the increase of pathogens or the increase of susceptibility of the contaminated species due to pollution for example.
Line 47: Subspecies name must be italicized.
Line 78: refrigerated . Necropsies
Remove space before .
Line 105-107: remove:
This section may be divided by subheadings. It should provide a concise and precise description of the experimental results, their interpretation, as well as the experimental conclusions that can be drawn.
LIne 164: remove the underline below the °
20ºC
Author Response
The paper is interesting and well written. My only concern is that there is no proof that these bacteria are cause of death of these animals.
The authors should be more cautious about the conclusions. For example line 17:
"Histological and molecular techniques were performed, to diagnose the aetiological agents involved in their mortality."
"Involved" is too strong. The authors should add "could be involved".
We have changed it accordingly
Minor editing problems:
line 19: indicated the presence of P. damselae subsp damselae y Photobacterium swingsii in the analyzed
Subspecies name must be italicized and change "y" by "and".
We have changed them accordingly
Line 33: criteria 1 (I, IV)
Please explain, because UICN red list criteria are not indicated like this:
See https://portals.iucn.org/library/node/7977
We have explained the Balearic Red list criteria in the manuscript
Line 40: with the increase of pathogens in marine water,
It can be the increase of pathogens or the increase of susceptibility of the contaminated species due to pollution for example.
We have changed it accordingly and we added a new sentence and references
Line 47: Subspecies name must be italicized.
We have changed it accordingly
Line 78: refrigerated . Necropsies
Remove space before .
We have changed it accordingly
Line 105-107: remove:
This section may be divided by subheadings. It should provide a concise and precise description of the experimental results, their interpretation, as well as the experimental conclusions that can be drawn.
We have changed it accordingly
LIne 164: remove the underline below the °
20ºC
We have changed it accordingly
Round 2
Reviewer 1 Report
I did not do any correction comment because, in my opinion, your work presents a big methodology problem.
I don't know for Photobacterium swingsii, but Photobacterium damselae might be naturally present in fish's intestine, and the same can happen with some shark species.
Therefore, as you mixed the anterior intestine with the other organs for DNA extraction, the Photobacterium damselae present in the intestine might have contaminated the organs mixture. This means that you might have detected the Photobacterium damselae from the intestine (which might have been normal) and not the real mortality cause.
The fact that you detected two different Photobacterium species is a bit suspicious. It is possible that the true mortality cause might have been P. swingsii and not P. damselae.
To conclude, you should not mix the organs. It would be preferable to analyze every organs except intestine and stomach (in this case mixing organs could be ok), or including intestine and stomach but separately.
Reviewer 2 Report
The review points has been addressed and recommended for publication